# Association of Sugar-Sweetened Beverage Consumption and Moderate-to-Vigorous Physical Activity with Childhood and Adolescent Overweight/Obesity: Findings from a Surveillance Project in Jiangsu Province of China

**DOI:** 10.3390/nu15194164

**Published:** 2023-09-27

**Authors:** Jinxia Yu, Feng Huang, Xiyan Zhang, Hui Xue, Xiaoyan Ni, Jie Yang, Zhiyong Zou, Wei Du

**Affiliations:** 1School of Public Health, Southeast University, Nanjing 210009, China; 230229477@seu.edu.cn (J.Y.); 220203796@seu.edu.cn (H.X.); 220223619@seu.edu.cn (X.N.); 2Fujian Provincial Center for Disease Control and Prevention, Fuzhou 350001, China; 220203799@seu.edu.cn; 3Department of Child and Adolescent Health Promotion, Jiangsu Provincial Center for Disease Control and Prevention, Nanjing 210009, China; xiyanzhang0220@jscdc.cn; 4Institute of Child and Adolescent Health, School of Public Health, Peking University, Beijing 100191, China

**Keywords:** sugar-sweetened beverage, children and adolescents, overweight/obesity, school age, moderate-to-vigorous physical activity

## Abstract

Sugar-sweetened beverage (SSB) consumption and inadequate moderate-to-vigorous physical activity (MVPA) have been suggested as potential contributors to overweight/obesity during childhood or adolescence; however, the results of previous studies are inconsistent. It was crucial to estimate the independent and joint association of SSB consumption and inadequate MVPA for childhood and adolescent overweight/obesity. The “Surveillance for Common Disease and Health Risk Factors Among Students in Jiangsu Province 2021–2022” initiative provided us with representative population-based data that we studied. SSB consumption and inadequate MVPA were determined by self-reported SSB habit and MVPA frequency (days/week). The body mass index for each gender and age subgroup was used to identify those who were overweight or obese. With stratified analyses to ascertain differences in age or gender, we employed the logistic regression model to assess the association of SSB and MVPA with overweight/obesity and applied the likelihood ratio test to explore the interactions. Approximately 38.2% of the study population (119,467 students aged 8–17) were overweight/obese. After adjusting covariates, SSB consumption or inadequate MVPA was associated with overweight/obesity (OR = 1.05, 95% CI = 1.02–1.07; and OR = 1.07, 95% CI = 1.03–1.10). In comparison to students with “no SSB consumption and adequate MVPA”, those with “SSB consumption and inadequate MVPA” had a higher risk of being overweight/obese (OR = 1.13, 95% CI = 1.08–1.18). Regardless of age and gender subgroups, the correlation of SSB and MVPA alone and together with being overweight/obese was generally similar, with the adolescent group aged 13–17 years (OR = 1.15, 95% CI = 1.09–1.22) and females (OR = 1.09, 95% CI = 1.02–1.17) being more susceptible. Moreover, there was a significant interaction between SSB consumption and gender (*p* < 0.001), as well as between SSB consumption and inadequate MVPA (*p* = 0.008). Hence, SSB consumption in students is significantly associated with overweight/obesity, especially when MVPA is inadequate. In light of the rapidly expanding childhood and adolescent obesity epidemic, proper attention should be given to these modifiable behaviors, particularly SSB and MVPA.

## 1. Introduction

Childhood and adolescent overweight or obesity has emerged as an alarming health issue in China and around the world, with the prevalence of obesity approximately eight times higher globally than that of forty years ago [1]. There was an increase in childhood and adolescent (2–19 years old) obesity prevalence in America from 10% in 1988–1994 to 19.2% in 2017–2018 [2]. Although obesity was formerly only found in high-income nations, it has become increasingly widespread in countries with low and middle incomes. On the basis of a *Lancet* study, 6.8% and 3.6% were the rates of overweight and obesity, respectively, among Chinese children (under 6 years old) in 2015–2019, while children and teenagers aged 6–17 during the same period had rates of 11.1% and 7.9% [3]. The severity of China’s childhood obesity crisis has also increased as a result of the coronavirus disease 2019 pandemic in recent years [4]. Childhood and adolescent obesity might raise the risk of developing hypertension, hyperlipidemia, impaired glucose metabolism, and obstructive sleep apnea [5], and it is highly associated with obesity in adulthood [6]. Thus, obesity-related diseases can place a heavy burden on health care and have serious social and economic ramifications [7]. The direct financial costs of obesity-related chronic diseases in China are expected to increase to RMB 49.05 billion per year by 2030 [8]. Therefore, confirming the key obesity and overweight risk factors is crucial, as is developing effective intervention strategies.

Among all known risk factors for overweight/obesity, sugar-sweetened beverage (SSB) consumption and inadequate moderate-to-vigorous physical activity (MVPA) were particularly prevalent during childhood or adolescent, which seemed to be a growing issue in the student populations [9,10]. In the National Health and Nutrition Examination Survey 2011–2014, 62.9% of American youths aged 2–19 years consumed at least one SSB per day, and the daily energy intake from SSB was 143 kcal, accounting for 7.3% of total daily energy [11]. China produced 177.6 million tons of SSB annually in 2019, and children aged 8 to 14 consumed more SSB, going from 329 mL/day in 1998 to 715 mL/day in 2008 [12]. There has been abundant literature exploring the role of SSB consumption in overweight/obesity specifically for children and adolescents [13,14,15,16,17,18,19], yet the findings are inconsistent [20,21,22]. Meanwhile, few researchers have paid attention to the association between SSB consumption and overweight/obesity in Chinese children and adolescents. There is proof that general physical activity, especially MVPA, has a substantial negative correlation with body mass index (BMI) [23,24,25,26,27]. Although the World Health Organization (WHO) recommends that youths aged 5–17 years should perform 60 min of MVPA daily [28], the proportion of children and adolescents meeting this criterion is quite low, indicating a general lack of MVPA among Chinese schoolchildren [29]. Considering that SSB consumption and inadequate MVPA are commonly co-occurring in children and adolescents, knowing their joint association is essential for developing healthy strategies to prevent overweight/obesity. An increasing corpus of literature demonstrates that risk factor combinations could influence healthy outcome in diverse ways that may not be explainable by analyzing the influences of each risk factor alone [30,31,32]. The purposes of this study were to explore the independent and joint relationships of SSB consumption and inadequate MVPA with being overweight/obese for Chinese children and adolescents, taking into account whether those associations would be altered in age or sex subgroup analyses.

## 2. Materials and Methods

### 2.1. Study Population

The data for this paper were obtained from the “Surveillance for Common Disease and Health Risk Factors Among Students in Jiangsu Province 2021–2022” by employing a cluster random sampling strategy to select 168,842 participants in Jiangsu Province of China. A self-administered questionnaire was given to each participant to complete. There have been several descriptions of this surveillance project’s specifics [33]. Some participants in the study were excluded since they were too young to comprehend and respond to the health risk factor assessment (*n* = 24,862), college students over 17 years of age were not well represented for the wider 17-year-old population (*n* = 12,559), or because their data regarding the analyzed variables were lacking (*n* = 11,954). In total, 119,467 students from 8 to 17 years old were finally included in this research (Figure 1). The study protocol was granted exemption by the Ethics Committee of Jiangsu Provincial Center for Disease Prevention and Control. All participants provided informed consent.

### 2.2. Measures

#### 2.2.1. Childhood and Adolescent Overweight/Obesity

Depending on a standard established by the China Obesity Task Force, a child or adolescent was deemed overweight or obese whenever their BMI reached or exceeded the 85th/95th centile for their specified gender and age [34]. The participants’ height (cm) and weight (kg) were assessed by experienced investigators using standardized devices and procedures, and refined to the nearest 0.1 unit. Afterwards, we obtained the BMI from division of weight (kg) into height (m) squared. 

#### 2.2.2. SSB Consumption and Inadequate MVPA

SSB habit was evaluated by asking the respondents “How many times in the last week have you consumed SSB?” with three answer choices, that is, “Never SSB consumption”, “SSB consumption less than once a day”, and “SSB consumption once a day or more”. Respondents who never consumed SSB were classified as non-SSB consumers, while the others were classified as SSB consumers.

MVPA was evaluated by asking the respondents “How many days in the last week did you exercise that caused you to breathe heavily and have a rapid heartbeat for at least 60 min?” Children and teenagers aged 5 to 17 ought to undertake MVPA for a minimum of 60 min on average daily, according to the World Health Organization’s (2020) recommendations [28]; hence, inadequate MVPA was defined as when children and adolescents self-reported that they practiced MVPA for a minimum of 1 h (60 min) daily, fewer than seven days weekly.

#### 2.2.3. Covariates

Demographic variables such as age, gender, regions, residence, family types, and level of education were obtained from the self-reported questionnaire [33]. Age group was classified into 8–12 and 13–17 years. Gender was categorized as male and female. The provincial regions were divided into the southern, central, and northern parts of Jiangsu. Residence was categorized as urban and rural. Family types were divided into core families and others, such as extended families, one-parent families, re-married families, and inter-generational families. Paternal education and maternal education were classified into four categories: primary or illiterate, junior or senior high school, college, or post-graduate and beyond. 

### 2.3. Statistics

Descriptive analysis of demographic information on categorical variables was conducted utilizing numbers and proportions, while the differences between these characteristic variables and the body-weight status were verified by chi-squared testing. For the assessment of the independent association for SSB consumption or inadequate MVPA and overweight/obesity, multiple logistic regressions were performed. Age variable and gender variable were controlled for in Model 1; residence, regions of the province, family types, and paternal education and maternal education were also controlled for in Model 2; and inadequate MVPA or SSB consumption was adjusted for in Model 3. 

To establish the joint association of SSB consumption and inadequate MVPA on overweight/obesity, participants were classified as four groups depending on the combination of the two variables: (1) students without SSB consumption and meeting MVPA recommendations (henceforth termed “no SSB consumption + adequate MVPA”); (2) students without SSB consumption and not meeting MVPA recommendations (henceforth termed “no SSB consumption + inadequate MVPA”); (3) students consuming SSB and meeting MVPA recommendations (henceforth termed “SSB consumption + adequate MVPA”); and (4) students consuming SSB and not meeting MVPA recommendations (henceforth termed “SSB consumption + inadequate MVPA”). Subsequently, multivariate logistic regressions were applied to model the joint associations among SSB-MVPA and overweight/obesity.

The stratification variable was removed from the model in order to conduct a further analysis of age and gender subgroup stratification using the same modeling approach as previously mentioned. To determine whether significant interactions existed for age or gender, likelihood ratio tests for models having and not having an interactive term were used.

For the robustness of our results, there were additionally several types of sensitivity analyses performed in this paper. We investigated the independent association of SSB habit (considered as a triple categorical factor: never (never SSB consumption), sometimes (SSB consumption less than once a day), and always (SSB consumption once a day or more)) and MVPA level (considered as a tri-categorical variable: low frequency (0 to 3 days weekly), moderate frequency (4 to 5 days weekly), and high frequency (6 to 7 days weekly)), according to prior research [29] on overweight/obesity. In addition, we examined the independent and joint associations of SSB consumption and inadequate MVPA with consecutive variables of BMI. Statistical *p*-values were two-sided analyses, where a *p*-value under 0.05 was defined as statistically significant. Utilizing Stata version 16.0 (Stata Corporation, College Station, TX, USA), all statistical analyses were carried out.

## 3. Results

### 3.1. Participant Characteristics

Overall, 119,467 subjects were enrolled onto our study (Figure 1), comprising 66,471 students aged 13–17 (55.6%) and 52,996 students aged 8–12 (44.4%). In total, 52.5% of the study participants were male, while 47.5% were female. The prevalence of overweight/obesity was 38.2% overall, with a slightly higher percentage of students with abnormal weight in the 8–12 age group compared to the 13–17 group (40.1% vs. 36.7%, respectively), and a slightly higher percentage of male students than female students (45.4% vs. 30.2%, respectively) (Appendix A). Appendix A shows that most of the participants (63.7%) claimed “SSB consumption and inadequate MVPA”, followed by 21.9% having “no SSB consumption and inadequate MVPA”, 9.5% having “SSB consumption and adequate MVPA”, and 4.9% having “no SSB consumption and adequate MVPA”. A relatively larger proportion of SSB consumption and inadequate MVPA was found in teenagers aged 13 to 17 (70.4% vs. 55.2%), particularly in females (65.0% vs. 62.5%).

The demographic information of this study is presented in Table 1. Students aged 8–12 years old, males, those living in Jiangsu Province’s central regions, those coming from non-core families, or those with higher maternal education levels were at heightened risk of becoming overweight or obese. In the Appendix A, the characteristics of participants categorized according to SSB consumption, inadequate MVPA, or joint SSB-MVPA are additionally shown in Appendix A.

### 3.2. Independent Association with Overweight/Obesity in SSB or MVPA

As shown in Table 2, participants who reported consuming SSB tended to have a higher probability of being overweight/obese compared to those not consuming SSB (Model 1, Model 2, and Model 3: OR = 1.05, 95% CI = 1.02–1.07). In contrast to adequate MVPA, inadequate MVPA was also independently associated with a higher risk of being overweight or obese (Model 1: OR = 1.07, 95% CI = 1.03–1.11; Model 2: OR = 1.06, 95% = 1.03–1.10; Model 3: OR = 1.07, 95% CI = 1.03–1.10). In terms of the association with overweight/obesity, there was a significant interaction between SSB consumption and gender (*p* < 0.001), as well as between SSB consumption and inadequate MVPA (this result is not presented in Table 2, *p* = 0.008).

### 3.3. Joint Association with Overweight/Obesity in SSB or MVPA

As opposed to students who had “no SSB consumption and adequate MVPA”, those who had “SSB consumption and adequate MVPA” (Model 1 of Table 3: OR = 1.03, 95% CI = 1.00–1.06; Model 2 of Table 3: OR = 1.03, 95% CI = 1.00–1.06) and “SSB consumption and inadequate MVPA” (Model 1 of Table 3: OR = 1.13, 95% CI = 1.08–1.18; Model 2 of Table 3: OR = 1.13, 95% CI = 1.08–1.18) possessed higher risks of being overweight or obese.

### 3.4. Subgroup Analysis of Age and Gender Stratification

In the age subgroup analysis, the statistically significant association of SSB-overweight/obesity was observable only among adolescents of 13–17 years old, in which a similar result was found for MVPA-overweight/obesity (Table 2). In the gender subgroup analysis, statistically significant effects of SSB-overweight/obesity were observed only in females, while statistically significant effects of MVPA-overweight/obesity were found only in males (Table 2). Regarding the stratified analysis of the SSB-MVPA joint classification and overweight/obesity correlation (Figure 2), the findings were supportive of a consistently higher risk in stratified subgroups, where the joint association appeared to be more prominent for the age group of 13–17 years and among females.

### 3.5. Sensitivity Analysis

The independent relationships of SSB habit (in Appendix A, this is recognized as a triple categorical factor) and MVPA level (in Appendix A, this is recognized as a triple categorical factor) with being overweight/obese were further investigated in our study. Appendix A reveals that compared to students who had never consumed SSB, those who “sometimes” consumed SSB were significantly associated with a raised risk of being overweight/obese. A similar association was also observable among students who “always” consumed SSB, albeit the findings were not of statistical significance. Appendix A shows that students with moderate frequency and low frequency of MVPA both had an elevated risk of overweight/obese compared to those with high frequency of MVPA.

Furthermore, employing linear regressions, we investigated the independent and joint associations between SSB consumption or inadequate MVPA and the continuous outcome variable BMI, as presented in Appendix A. These findings were mostly in accordance with our primary results, indicating that both SSB consumption and inadequate MVPA were independently associated with an enhanced risk of elevated BMI, and that students having “SSB consumption and inadequate MVPA” were at the highest risk of increased BMI compared to other groups (Appendix A). Stratified analyses additionally revealed that students in the 13–17 age group and females performed more prominently in the aforementioned associations, albeit with slight differences from the main results (Appendix A).

## 4. Discussion

Overweight/obesity refers to a chronic metabolic disease where energy ingestion exceeds energy consumption under the interaction of genetic and circumstantial factors, resulting in excessive accumulation of body fat and jeopardizing health. Approximately 38.2% of the children and adolescents aged 8–17 years in this survey were overweight/obesity, which was close to the survey outcomes related to the physically healthy status of students in Jiangsu Province in 2017–2019 (32.9%) and 2019–2020 (33.2%) [30,33], and higher than the results of the 2017–2019 study of overweight/obese children aged 8–13 years in northern Chinese cities (18.8%–23.6%) [35]. This may be due to the fact that Jiangsu Province is one of the economically developed provinces along the eastern coast of China, where different socioeconomic and regional environmental characteristics aggravate the heightened risks of being overweight/obese [36].

There was a direct association (independent of inadequate MVPA) between SSB consumption and overweight/obesity during childhood and adolescence in our study, in agreement with findings from an Irish study of 1075 students aged 8–11 years in 2012 [17]. Similar to our results, the 2013–2015 Hellenic National Nutrition and Health Survey (HNNHS) showed that SSB consumption in 1165 youth aged 2–18 years increased the probability for overweight/obesity [37]. Furthermore, similar findings were obtained in a Greek study revealing that SSB consumption was related to visceral obesity in 2665 schoolchildren aged 9–13 years in 2007–2009 [38]. This association between SSB consumption and overweight/obesity was possibly due in part to the high sugar content in SSB, which is low in satiety and potentially incompletely compensates for total energy, contributing to greater energy consumption [39,40]. However, our findings were different from those of Valente et al., who did not observe an association between SSB consumption and an elevated risk of overweight among 1675 Portuguese schoolchildren aged 5–10 years in 2010 [41]. In another study of 268 children (aged 10 years) in three rural states of the western United States from 2001 to 2003, there was no significant association between SSB consumption and BMI [42]. These inconsistencies could result from variations in study sample sizes, obesity assessment methods, and demographic factors. Additionally, we discovered a direct association (independent of SSB consumption) between inadequate MVPA and overweight/obesity, which was supported by an Australian survey where inadequate MVPA increased the prevalence of overweight/obesity among 7908 adults in 2002–2008 [43]. The 2015 Ontario Student Drug Use and Health Survey (OSDUHS) also demonstrated a significant association between inadequate MVPA and overweight/obesity in 9866 schoolchildren aged 11–17 years [44]. One argument was that people with adequate MVPA might have more vulnerability to a system of appetitive control through improvements in compensation regulation of food densities and energy contents [45]. Moreover, physical activity tends to influence appetite by lowering the levels of various hormones in the body, which may promote changes in the consumption of foods and beverages following exercise. A previous article has highlighted the effectiveness of therapies such as cardio-exercise on indicators of insulin resistance in young obese people [46].

In this research, it became clear that students who consumed SSB and had inadequate MVPA were at higher risk of becoming overweight or obese than those who did not consume SSB or had adequate MVPA, when SSB and MVPA were examined jointly. There is a paucity of research on the joint associations of SSB consumption and inadequate MVPA with obesity, yet numerous studies have demonstrated that building healthy behaviors in diet and physical activities throughout childhood and adolescence should be an essential measure to prevent obesity along with non-communicable diseases [47,48]. Although this paper cannot substantiate any exact physiological mechanism to explain the current findings, we hypothesize that students with SSB consumption and inadequate MVPA were more susceptible to the development of obesity-related health risk factors and diseases than those with less unhealthy behaviors. Meanwhile, we found a significant interaction between SSB consumption and gender, where a rational explanation may be that gender factors could influence the awareness of health and body image management, resulting in a reduction in SSB consumption [49]. The significant interaction between SSB consumption and inadequate MVPA was also observed, which may be due to the frequent exposure of children and adolescents to advertisements for sugar-sweetened beverages promoting exercise [50].

In addition, this paper evaluated the association of age and gender with SSB or MVPA and overweight/obesity. It was observed that female students seemed to be more prone to SSB consumption alone or in combination with inadequate MVPA than their counterparts. A systematic review suggested that compared to boys, girls may be more vulnerable to obesity-related effects of adverse childhood experiences due to their relational sensitivity [51], which may be an important reason for the gender differences in the association of SSB with obesity. We found that the group of 13–17-year-olds seemed to be more sensitive than their counterparts to consuming SSB alone, perhaps due to the fact that adolescents consumed SSB as a coping mechanism under academic stress [52,53]. Similarly, when we observed the joint association of SSB consumption and inadequate MVPA, we noted that the 13–17 age group continued to show more vulnerability, a potential explanation being that adolescents were more physically inactive, thus exacerbating the implications of SSB [54]. Variations in physiology or hormones may account for age and sex variances between SSB and MVPA in relation to overweight/obesity; the specific mechanisms behind these differences should be studied in additional research.

Our research assessed the independent and joint associations of SSB consumption and inadequate MVPA with overweight/obesity in children and adolescents, identifying the differences in age or gender and the interactions. Therefore, we recommend that in future overweight/obesity prevention efforts, gender- and age-specific personalized interventions could be implemented to reduce SSB consumption and increase MVPA, and advocate for authorities to limit advertising of sugar-sweetened beverages promoting exercise. This study utilized newly available data from children and adolescents in Jiangsu with a large sample size, and the results will provide policymakers and youth health advocates with informative recommendations for considering appropriate responses to reduce childhood and adolescent overweight/obesity risk. Despite this, there are some limitations in this paper as well. First, given that the majority of the variables in this study were assessed using self-reported data, recall bias may be present. Consequently, caution is urged when interpreting the results. Second, these cross-sectional surveillance data limit our confirmation of a causal association between SSB or MVPA and overweight/obesity in childhood and adolescence. Future investigations using longitudinal study data are warranted. Third, this investigation also lacked more comprehensive information about SSB and MVPA, such as food species or energy density of SSB, duration or quality of MVPA, etc. Fourth, this study did not include any participants younger than 8 years old. Fifth, other aspects of dietary intake were not included in our analyses owing to non-measurement. Lastly, even accounting for a large number of potential factors, confounding bias may still be present due to unobserved traits.

## 5. Conclusions

This study showed that SSB consumption and inadequate MVPA were independently and jointly associated with increased risk of being overweight/obese in school-aged students. The joint association also appeared to be more pronounced in the group of 13–17-year-olds and in females. Additional investigations are warranted to clarify the exact mechanism lying behind these associations, specifically extensive cohort studies in different populations. In summary, children and adolescents require more interventions to reduce their intake of SSB and increase their physical activity behaviors, as these could help them maintain a normal weight. Accordingly, greater initiatives aimed at changing these modifiable behaviors are advised as a response to the global epidemic of childhood and teenage obesity.

## Figures and Tables

**Figure 1 nutrients-15-04164-f001:**
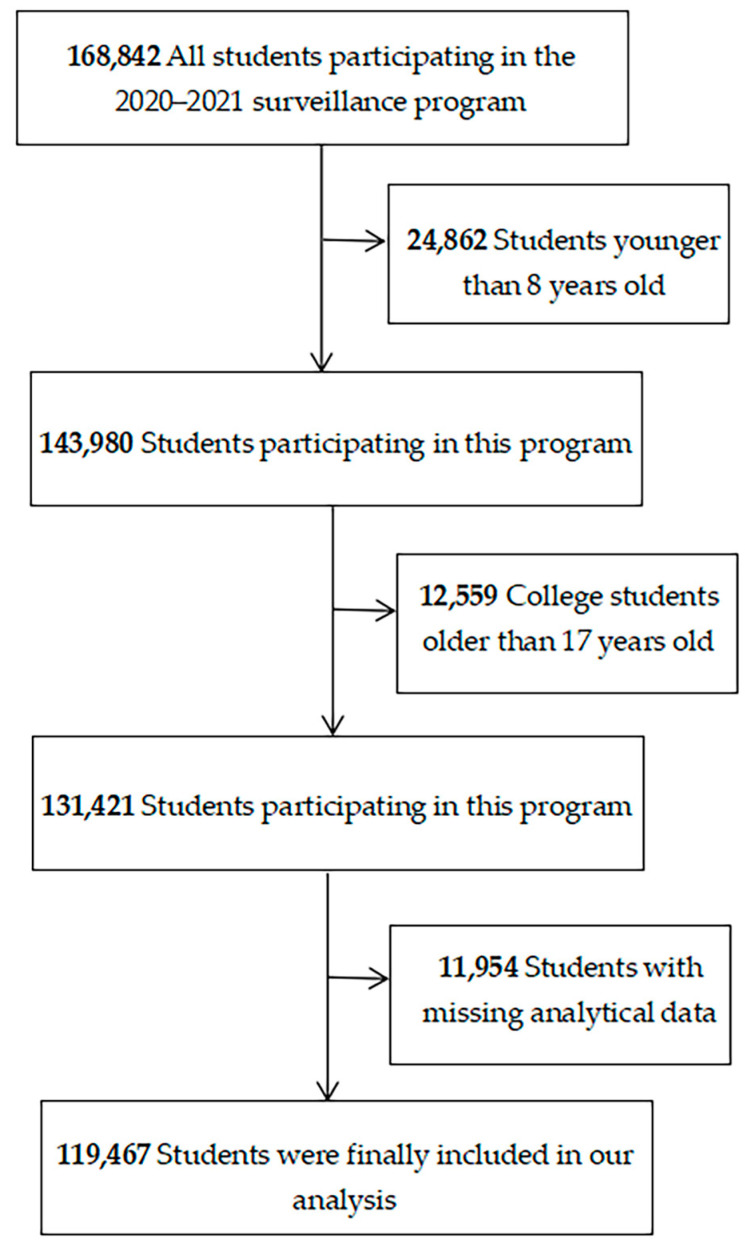
Flow chart of the population included in this study.

**Figure 2 nutrients-15-04164-f002:**
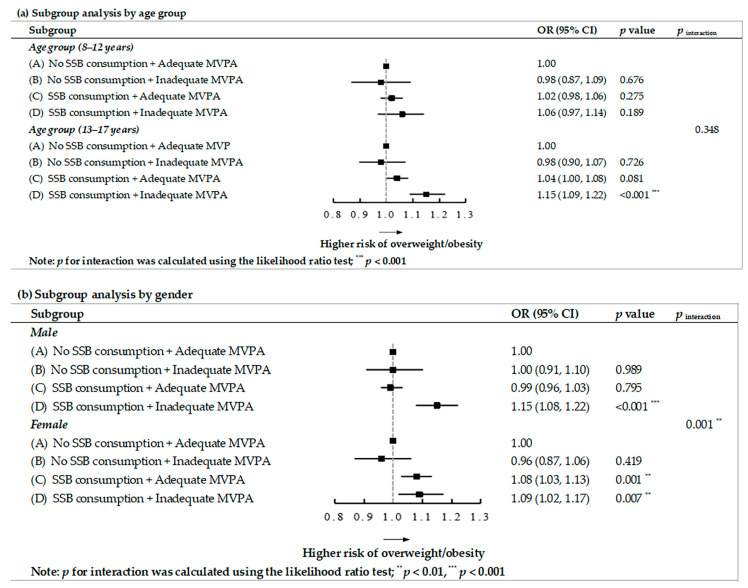
The joint association of SSB and MVPA with overweight/obesity in stratified subgroups of age or gender.

**Table 1 nutrients-15-04164-t001:** The demographic information of 119,467 students based on body weight status in Jiangsu Province of China.

Characteristics	Normal (73,805)	Overweight/Obesity (45,662)	*p*-Value ^a^
	*n* (%)		
**Age group, year**			
8–12	31,724 (59.86)	21,272 (40.14)	<0.001 ***
13–17	42,081 (63.31)	24,390 (36.69)	
**Gender**			
Male	34,232 (54.55)	28,527 (45.45)	<0.001 ***
Female	39,573 (69.78)	17,135 (30.22)	
**Residence**			
Urban	42,058 (61.70)	26,110 (38.30)	0.507
Rural	31,747 (61.89)	19,552 (38.11)	
**Region of Jiangsu Province**			
Southern	32,083 (63.05)	18,804 (36.95)	<0.001 ***
Central	10,305 (59.48)	7020 (40.52)	
Northern	31,417 (61.30)	19,838 (38.70)	
**Family types**			
Core families	33,085 (62.81)	19,587 (37.19)	<0.001 ***
Others ^b^	40,720 (60.96)	26,075 (39.04)	
**Maternal education**			
Primary or illiterate	8742 (62.43)	5262 (37.57)	0.036 *
Junior or senior high school	47,738 (61.87)	29,420 (38.13)	
College	15,408 (61.35)	9707 (38.65)	
Post-graduate and beyond	1917 (60.09)	1273 (39.91)	
**Paternal education**			
Primary or illiterate	4814 (62.07)	2942 (37.93)	0.493
Junior or senior high school	49,963 (61.88)	30,775 (38.12)	
College	16,798 (61.48)	10,525 (38.52)	
Post-graduate and beyond	2230 (61.10)	1420 (38.90)	

^a^ Chi-square testing was employed to calculate *p*-values. ^b^ Comprising extended families, one-parent families, re-married families, and inter-generational families. * *p* < 0.05; *** *p* < 0.001.

**Table 2 nutrients-15-04164-t002:** The independent association of SSB and MVPA with overweight/obesity in overall population and subgroups.

Variables	Overall	Age Subgroup			Gender Subgroup		
	OR (95% CI)	(8–12) OR (95% CI)	(13–17) OR (95% CI)	*p_i_* ^d^	Male OR (95% CI)	Female OR (95% CI)	*p_i_* ^d^
SSB Consumption (yes vs. no)							
Model 1 ^a^	1.05 (1.02, 1.07) **	1.03 (0.99, 1.07)	1.06 (1.02, 1.10) **	0.409	1.01 (0.97, 1.05)	1.09 (1.04, 1.13) ***	<0.001 ***
Model 2 ^b^	1.05 (1.02, 1.07) **	1.03 (0.99, 1.07)	1.06 (1.02, 1.10) **	0.426	1.01 (0.98, 1.05)	1.09 (1.04, 1.13) ***	<0.001 ***
Model 3 ^c^	1.05 (1.02, 1.07) **	1.03 (0.99, 1.07)	1.06 (1.02, 1.10) **	0.418	1.01 (0.98, 1.05)	1.09 (1.04, 1.13) ***	<0.001 ***
Inadequate MVPA (yes vs. no)							
Model 1 ^a^	1.07 (1.03, 1.11) ***	1.02 (0.95, 1.08)	1.08 (1.04, 1.13) ***	0.052	1.11 (1.06, 1.16) ***	1.01 (0.96, 1.06)	0.244
Model 2 ^b^	1.06 (1.03, 1.10) ***	1.01 (0.95, 1.08)	1.08 (1.03, 1.12) ***	0.064	1.11 (1.06, 1.17) ***	1.00 (0.95, 1.05)	0.266
Model 3 ^c^	1.07 (1.03, 1.10) ***	1.01 (0.95, 1.08)	1.08 (1.03, 1.12) ***	0.064	1.11 (1.06, 1.17) ***	1.00 (0.95, 1.05)	0.258

^a^ Age variable and gender variable were controlled for in Model 1. ^b^ Based on Model 1, Model 2 also controlled for residence, regions of the province, family types, and paternal education and maternal education. ^c^ Based on Model 2, Model 3 also controlled for inadequate MVPA or SSB consumption. ^d^ Comparison of the two models with and without the interaction element led to the computation of *p_i_* using the likelihood ratio test. Abbreviations: odds ratio (OR), confidence interval (CI). ** *p* < 0.01, *** *p* < 0.001.

**Table 3 nutrients-15-04164-t003:** The joint association of SSB and MVPA with overweight/obesity in the total study population.

Groups	Model 1 ^a^		Model 2 ^b^	
	OR (95% CI)	*p* Value	OR (95% CI)	*p* Value
(A) No SSB consumption + Adequate MVPA	1.00		1.00	
(B) No SSB consumption + Inadequate MVPA	0.99 (0.92, 1.06)	0.684	0.98 (0.92, 1.05)	0.634
(C) SSB consumption + Adequate MVPA	1.03 (1.00, 1.06)	0.041 *	1.03 (1.00, 1.06)	0.038 *
(D) SSB consumption + Inadequate MVPA	1.13 (1.08, 1.18)	<0.001 ***	1.13 (1.08, 1.18)	<0.001 ***

^a^ Age variable and gender variable were controlled for in Model 1. ^b^ Based on Model 1, Model 2 also controlled for residence, regions of the province, family types, and paternal education and maternal education. Abbreviations: odds ratio (OR), confidence interval (CI). * *p* < 0.05, *** *p* < 0.001.

## Data Availability

Data are not available as the participants in our research disagreed with the public sharing of their data.

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
