# Peer review of "Association of Sugar-Sweetened Beverage Consumption and Moderate-to-Vigorous Physical Activity with Childhood and Adolescent Overweight/Obesity: Findings from a Surveillance Project in Jiangsu Province of China"

_nutrients, 2023, doi:10.3390/nu15194164_

Round 1

Reviewer 1 Report

Thank you for the opportunity to review this interesting manuscript.

1.       The authors should be careful to identify the context (i.e. the country, date and sample) of studies reported from elsewhere because the findings from a study of Chinese children and adolescents may not be generalisable to other populations. This is particularly the case when only one component of an unhealthy diet is included in analyses, and it is measured crudely (i.e. self-report into two categories – ever consume or never consume).   

2.       The potential for interaction is included in the analyses and found to be statistically significant in many models. However which interactions were investigated and the meaning of the findings are not made clear.

3.       The authors state (lines 304-305) that ‘the results provided policymakers and youth health advocates with informative recommendations for considering appropriate responses to reduce childhood and adolescent overweight/obesity risk’ If this were true, the recommendations should be stated. Are they different to recommending interventions to reduce SSB consumption and increase MVPA? Perhaps these recommendations could have been made prior to this report - what does this study contribute?

4.       Another weakness for the study (to be added) is that no other aspects of dietary intake were included in the analysis (they may not have been measured). It would be unlikely that other aspects of dietary intake are uniform across SSB consumers and non-consumers.

Minor points:

Abstract – the sample number in the analysis should be given (i.e. 119467 from age 8-17 years)

-          Is there statistical support for older age/female being more susceptible to SSB and lack of MVPA (line 31)?

-          I question the description of the association of SSB as a strong association with overweight/obesity (line 32). An odds ratio of 1.13 is not a strong association.

-          Are the odds ratios stated (lines 27-29) crude or adjusted for other factors?

-          OR should be singular in line 29

Introduction

Line 41 – 2013-14 figures for the US are about 10 years old – are there more recent figures?

Lines 45-47 – please provide the dates that the prevalence figures apply to.

Line 60 – the correct name of the survey is the National Health and Nutrition Examination Survey

Line 61-62 –the country that the statistics apply to should be identified.

Line 70-71 –the stated recommendation should be referenced to its source. It seems incomplete – is the daily duration part of the recommendation?

Line 90 – please clarify the reason college students over the age of 17 years were removed – was it because of a poor response rate to the survey, or because college students are not a good representation of the wider 17 year old population?

Line 100, 114 – the cited references (i.e. reference 33, 34, 35) appear to be to reports of other studies that used the same reference values. Please cite the primary source of the reference values.

Line 149-150 and Table 2 – The interactive term needs to be identified in both places for clarity – is it SSB*MVPA?

Discussion:

Lines 250-252: Please include the date of measurement for the comparison surveys (rather than state ‘in recent years’.

Line 256 – it is unclear what ‘individually linked’ means – would ‘associated with’ be a better term?

Lines 258-261 – prior research studies need to be described in more detail (i.e. references 18, 38, 39, 42, 43). It is not clear what populations were measured, how large the populations were, when they were measured and some appear to be aggregations of studies – can some clarity be offered?

Line 263 – the term ‘individual association’ is used – is this intended to be ‘independent association’? If so, the authors should clarify what the association is independent of – perhaps the authors mean ‘…association between inadequate MVPA and overweight/obesity independent of intake of SSB…’? The findings of references 44 and 45 need to be explained so that the reader understands the point the authors are making.

Line 289 – it is important to outline how the result of the systematic review (reference 50) is relevant to the sample studied.  Are the authors suggesting that adverse childhood experiences could be an explanatory factor for the associations they found ?

There are no references to, or discussion of the meanings of the many statistically significant interactions found.

The English language is generally understandable but in many places the meaning is unclear. 

Author Response

Dear reviewer:

Thank you for your positive and constructive comments and suggestions on my manuscript. We have carefully considered the suggestion of Reviewer and make some changes. We have tried our best to improve and made some changes in the manuscript. According to your comments, the changes are highlighted in red in the font for easy inspection in the revised manuscript (Clean Version). We hope this revision can make our paper more acceptable. The revisions were addressed point by point below.

[General Comment 1] The authors should be careful to identify the context (i.e. the country, date and sample) of studies reported from elsewhere because the findings from a study of Chinese children and adolescents may not be generalisable to other populations. This is particularly the case when only one component of an unhealthy diet is included in analyses, and it is measured crudely (i.e. self-report into two categories – ever consume or never consume).

Response: We sincerely appreciate the valuable comments. We have checked the literature carefully and added more contextual information from different studies.We revised the sentence as follows:

“in agreement with findings from an Irish study of 1075 students aged 8-11 years in 2012 [17]. Similar to our results, the 2013-2015 Hellenic National Nutrition and Health Survey (HNNHS) showed that SSB consumption in 1165 youth aged 2-18 years increased the probability for overweight/obesity [37]. Furthermore, similar findings were obtained in a Greek research revealing that SSB consumption was related to visceral obesity in 2665 schoolchildren aged 9-13 years in 2007-2009 [38]” [Pg8, Ln271-277]

“However, our findings were different from that of Valente et al. who did not observe an association between SSB consumption and an elevated risk of overweight among 1675 Portuguese schoolchildren aged 5-10 years in 2010 [41]. In another study of 268 children (aged 10 years) in three rural states of the western United States from 2001-2003, there was no significant association between SSB consumption and BMI [42]. ” [Pg8, Ln280-285]

“ which was supported by an Australian survey where inadequate MVPA increased the prevalence of overweight/obesity among 7908 adults in 2002-2008 [43]. The 2015 Ontario Student Drug Use and Health Survey (OSDUHS) also demonstrated a significant association between inadequate MVPA and overweight/obesity in 9866 schoolchildren aged 11-17 years [44].” [Pg8, Ln288-292] 

[General Comment 2] The potential for interaction is included in the analyses and found to be statistically significant in many models. However which interactions were investigated and the meaning of the findings are not made clear.

Response: Thanks for your kind reminders. We revised the sentence as follows:

“we employed the logistic regression models to assess the association of SSB and MVPA with overweight/obesity and applied the likelihood ratio test to explore the interactions.” [Pg1, Ln25-27]

“Besides, there was a significant interaction between SSB consumption and gender (p<0.001), as well as between SSB consumption and inadequate MVPA (p=0.008). ” [Pg1, Ln35-36]

“Meanwhile, we found a significant interaction between SSB consumption and gender, where a rational explanation may be that gender factors could influence the awareness of health and body image management resulting in a reduction of SSB consumption [49]. The significant interaction between SSB consumption and inadequate MVPA was also observed, which may be due to the frequent exposure of children and adolescents to advertisements for sugar-sweetened beverages promoting exercise [50].” [Pg9, Ln311-317]

[General Comment 3] The authors state (lines 304-305) that ‘the results provided policymakers and youth health advocates with informative recommendations for considering appropriate responses to reduce childhood and adolescent overweight/obesity risk’ If this were true, the recommendations should be stated. Are they different to recommending interventions to reduce SSB consumption and increase MVPA? Perhaps these recommendations could have been made prior to this report - what does this study contribute?

Response: Thanks for your significant reminding. We revised the sentence as follows:

“Therefore, we recommend that in future overweight/obesity prevention efforts, gender- and age-specific personalized interventions could be implemented to reduce SSB consumption and increase MVPA, and advocate for authorities to limit advertising of sugar-sweetened beverages promoting exercise.” [Pg9, Ln336-340]

[General Comment 4] Another weakness for the study (to be added) is that no other aspects of dietary intake were included in the analysis (they may not have been measured). It would be unlikely that other aspects of dietary intake are uniform across SSB consumers and non-consumers.

Response: Thanks for your constructive comments. We revised the sentence as follows:

“Fifth, other aspects of dietary intake were not included in our analyses owing to non-measurement.” [Pg10, Ln353-354]

[Minor Comment 1] the sample number in the analysis should be given (i.e. 119467 from age 8-17 years)

Response: Thanks for your excellent suggestion. We revised the sentence as follows:

“Approximately 38.2% of the study population (119467 students ages 8-17) were overweight/obese.” [Pg1, Ln27-28]

[Minor Comment 2] Is there statistical support for older age/female being more susceptible to SSB and lack of MVPA (line 31)?

Response: Thanks for your kind reminders. We revised the sentence as follows:

“ with the adolescent group aged 13-17 years (OR= 1.15, 95%CI= 1.09-1.22) and females (OR= 1.09, 95%CI= 1.02-1.17) being more susceptible.” [Pg1, Ln33-35]

[Minor Comment 3] I question the description of the association of SSB as a strong association with overweight/obesity (line 32). An odds ratio of 1.13 is not a strong association.

Response: Thanks for your careful checks. We are sorry for the inaccurate description. We revised the sentence as follows:

“SSB consumption in students is significantly associated with overweight/obesity,” [Pg1, Ln37]

[Minor Comment 4] Are the odds ratios stated (lines 27-29) crude or adjusted for other factors?

Response: Actually, the stated odds ratios were adjusted for other factors. Thanks for your kind reminders. We revised the sentence as follows:

“After adjusting covariates, SSB consumption or inadequate MVPA was associated with overweight/obesity” [Pg1, Ln28-29]

[Minor Comment 5] OR should be singular in line 29

Response: Thanks for your significant reminding. We revised the sentence as follows:

“those with ‘SSB consumption and inadequate MVPA’ had a higher risk (OR=1.13, 95%CI= 1.08-1.18). ” [Pg1, Ln31]

[Minor Comment 6] Line 41 – 2013-14 figures for the US are about 10 years old – are there more recent figures?

Response: We sincerely appreciate the valuable comments. We did our best to add more recent figures. We revised the sentence as follows:

“There was an increase of childhood and adolescent (2-19 years old) obesity prevalence in America from 10% in 1988–1994 to 19.2% in 2017–2018” [Pg2, Ln46-48]

[Minor Comment 7] Lines 45-47 – please provide the dates that the prevalence figures apply to.

Response: Thanks for your excellent suggestion. We revised the sentence as follows:

“On the basis of a Lancet study, 6.8% and 3.6% were the rates of overweight and obesity, respectively, among Chinese children (under 6 years old) in 2015-2019, while kids and teenagers aged 6-17 during the same period had rates of 11.1% and 7.9%” [Pg2, Ln50-53]

[Minor Comment 8] Line 60 – the correct name of the survey is the National Health and Nutrition Examination Survey

Response: Thanks for your kind reminders. We revised the sentence as follows:

“In the National Health and Nutrition Examination Survey 2011-2014” [Pg2, Ln65-66]

[Minor Comment 9] Line 61-62 –the country that the statistics apply to should be identified.

Response: Thanks for your significant reminding. We revised the sentence as follows:

“ 62.9% of American youth aged 2-19 years consumed at least one SSB per day” [Pg2, Ln66-67]

[Minor Comment 10] Line 70-71 –the stated recommendation should be referenced to its source. It seems incomplete – is the daily duration part of the recommendation?

Response: We sincerely appreciate the valuable comments. We revised the sentence as follows:

“ Although the World Health Organization (WHO) recommends youth aged 5-17 years perform 60 minutes of MVPA daily [28]” [Pg2, Ln75-77]

[Minor Comment 11] Line 90 – please clarify the reason college students over the age of 17 years were removed – was it because of a poor response rate to the survey, or because college students are not a good representation of the wider 17 year old population?

Response: Thanks for your constructive comments. We revised the sentence as follows:

“ college students over 17 years of age were not well represented for the wider 17 year old population (n = 12559)” [Pg3, Ln97]

[Minor Comment 12] Line 100, 114 – the cited references (i.e. reference 33, 34, 35) appear to be to reports of other studies that used the same reference values. Please cite the primary source of the reference values.

Response: Thanks for your kind reminders. We have cited the primary source of the reference values. We revised the sentence as follows:

“28. World Health Organization. WHO guidelines on physical activity and sedentary behaviour world health organization. Https://apps.Who.Int/iris/bitstream/handle/10665/325147/who-nmh-pnd-2019.4-eng.Pdf?Sequence=1&isallowed=y%0ahttp://www.Who.Int/iris/handle/10665/311664%0ahttps://apps.Who.Int/iris/handle/10665/325147.” [Pg12, Ln466-468]

“34. Group of China Obesity Task Force. Body mass index reference norm for screening overweight and obesity in chinese children and adolescents. Https://rs.Yiigle.Com/cmaid/591042.” [Pg13, Ln481-482]

[Minor Comment 13] Line 149-150 and Table 2 – The interactive term needs to be identified in both places for clarity – is it SSB*MVPA?

Response: We apologized for not claiming clarity, actually the significant interactions in Table 2 were SSB intake and gender, while the result of SSB*MVPA was not presented in Table 2.We revised the sentence as follows:

“there was a significant interaction between SSB consumption and gender (p<0.001), as well as between SSB consumption and inadequate MVPA (this result was not presented in Table 2, p=0.008).” [Pg6, Ln202-204]

[Minor Comment 14] Lines 250-252: Please include the date of measurement for the comparison surveys (rather than state ‘in recent years’.

Response: Thanks for your significant reminding. We revised the sentence as follows:

“which was close to the survey outcomes related to the physically healthy status of students in Jiangsu Province in 2017-2019 (32.9%) and 2019-2020 (33.2%)” [Pg8, Ln263-264]

[Minor Comment 15] Line 256 – it is unclear what ‘individually linked’ means – would ‘associated with’ be a better term?

Response: We sincerely appreciate the valuable comments. We revised the sentence as follows:

“indicating both SSB consumption and inadequate MVPA were independently associated with an enhanced risk of elevated BMI” [Pg8, Ln251-252]

[Minor Comment 16] Lines 258-261 – prior research studies need to be described in more detail (i.e. references 18, 38, 39, 42, 43). It is not clear what populations were measured, how large the populations were, when they were measured and some appear to be aggregations of studies – can some clarity be offered?

Response: Thanks for your excellent suggestion. We revised the sentence as follows:

“in agreement with findings from an Irish study of 1075 students aged 8-11 years in 2012 [17]. Similar to our results, the 2013-2015 Hellenic National Nutrition and Health Survey (HNNHS) showed that SSB consumption in 1165 youth aged 2-18 years increased the probability for overweight/obesity [37]. Furthermore, similar findings were obtained in a Greek research revealing that SSB consumption was related to visceral obesity in 2665 schoolchildren aged 9-13 years in 2007-2009 [38]” [Pg8, Ln271-277]

“However, our findings were different from that of Valente et al. who did not observe an association between SSB consumption and an elevated risk of overweight among 1675 Portuguese schoolchildren aged 5-10 years in 2010 [41]. In another study of 268 children (aged 10 years) in three rural states of the western United States from 2001-2003, there was no significant association between SSB consumption and BMI [42]. ” [Pg8, Ln280-285]

[Minor Comment 17] Line 263 – the term ‘individual association’ is used – is this intended to be ‘independent association’? If so, the authors should clarify what the association is independent of – perhaps the authors mean ‘…association between inadequate MVPA and overweight/obesity independent of intake of SSB…’? The findings of references 44 and 45 need to be explained so that the reader understands the point the authors are making

Response: Thanks for your kind reminders. We revised the sentence as follows:

“There was a direct association (independent of inadequate MVPA) between SSB consumption and overweight/obesity during childhood and adolescence in our study,” [Pg8, Ln270-271]

“Additionally, we discovered a direct association (independent of SSB consumption) between inadequate MVPA and overweight/obesity,” [Pg8, Ln287-288]

“ which was supported by an Australian survey where inadequate MVPA increased the prevalence of overweight/obesity among 7908 adults in 2002-2008 [43]. The 2015 Ontario Student Drug Use and Health Survey (OSDUHS) also demonstrated a significant association between inadequate MVPA and overweight/obesity in 9866 schoolchildren aged 11-17 years [44].” [Pg8, Ln288-292] 

[Minor Comment 18] Line 289 – it is important to outline how the result of the systematic review (reference 50) is relevant to the sample studied. Are the authors suggesting that adverse childhood experiences could be an explanatory factor for the associations they found ?

Response: We sincerely appreciate the valuable comments. We revised the sentence as follows:

“A systematic review suggested that compared to boys, girls may be more vulnerable to obesity-related effects of adverse childhood experiences due to their relational sensitivity [51]” [Pg9, Ln321-323]

[Minor Comment 19] There are no references to, or discussion of the meanings of the many statistically significant interactions found.

Response: Thanks for your excellent suggestion. We revised the sentence as follows:

“ Meanwhile, we found a significant interaction between SSB consumption and gender, where a rational explanation may be that gender factors could influence the awareness of health and body image management resulting in a reduction of SSB consumption [49]. The significant interaction between SSB consumption and inadequate MVPA was also observed, which may be due to the frequent exposure of children and adolescents to advertisements for sugar-sweetened beverages promoting exercise [50].” [Pg9, Ln311-317]

Reviewer 2 Report

Dear Author,

The Manuscript Number nutrients-2621701, Association of Sugar-Sweetened Beverage Consumption and Moderate-to-Vigorous Physical Activity with Childhood and Adolescent Overweight/Obesity: Findings from a Surveillance Project in Jiangsu Province of China,  I consider is an engaging article that presents relevant information. However, it has some minor issues to be improved:

Results

Include the flow chart of population selection and the number of individuals at each stage of the study. 

The paper needs a revision of English grammar.

Author Response

Dear reviewer:
Thank you for your positive and constructive comments and suggestions on my manuscript. We 
have carefully considered the suggestion of Reviewer and make some changes. We have tried our 
best to improve and made some changes in the manuscript. According to your comments, the 
changes are highlighted in blue in the font for easy inspection in the revised manuscript (Clean 
Version). We hope this revision can make our paper more acceptable. The revisions were 
addressed point by point below.

[General Comment ] Include the flow chart of population selection and the number of individuals 
at each stage of the study. 
Response: We sincerely appreciate the valuable comments. We have added the flow chart of 
population selection and the number of individuals at each stage of the study.We revised the 
sentence as follows:
“In total, 119467 students from 8-17 years old were finally included into this research (Figure 1). ” 
[Pg3, Ln98-99] 

Figure 1. The flow chart of the population inclusion in this study. ” [Pg3, Ln101-102] 
“119467 subjects overall were enrolled in our study (Figure 1), 66471 students aged 13-17 (55.6%) 
and 52996 students aged 8-12 (44.4%). ” [Pg5, Ln173-174]
